# Three-Dimensional Cone-Beam Computed Tomography Evaluation of Changes in Naso-Maxillary Complex Associated with Rapid Palatal Expansion

**DOI:** 10.3390/diagnostics13071322

**Published:** 2023-04-02

**Authors:** Thomas Georgiadis, Christos Angelopoulos, Moschos A. Papadopoulos, Olga-Elpis Kolokitha

**Affiliations:** 1Orthodontist, Private Practice, 54621 Thessaloniki, Greece; 2Department of Oral Diagnosis and Radiology, School of Dentistry, National and Kapodistrian University of Athens, 11527 Athens, Greece; 3Department of Orthodontics, School of Health Sciences, Faculty of Dentistry, Aristotle University of Thessaloniki, 54124 Thessaloniki, Greece

**Keywords:** orthodontic diagnostics, cone-beam computed tomography, rapid palatal expansion, nasal airway, nasal cavity

## Abstract

The introduction of Cone-Beam Computed Tomography (CBCT) in orthodontics has added a new tool to diagnosis and treatment planning. The aim of this prospective clinical trial was to investigate the changes in the dimensions of the naso-maxillary complex in growing patients after RPE using CBCT. A total of 16 growing children (8 females, 6 males) with a mean age of 11, 12 ± 1 and 86 years underwent RPE as part of their comprehensive orthodontic treatment. CBCT scans were obtained before RPE (T1), immediately after RPE (T2) and 6 months after RPE (T3). The dimensions of the nasal width, nasal floor and the aperture of the midpalatal suture were calculated in different coronal slices of CBCT. Evaluation of the mean value variance per measurement at the three time intervals were performed using the paired Wilcoxon signed-rank test. Differences between the three time intervals were assessed by performing Multiple Pairwise Comparisons. A statistically significant increase in all measurements was seen immediately after RPE expansion (T2–T1) and six months after expansion (T3–T1). Between the end of expansion and 6 months in retention (T3–T2), a decrease was observed for all measurements. RPE can cause expansion of the nasal cavity in growing patients. The expansion of the midpalatal suture follows a triangular pattern of opening.

## 1. Introduction

Rapid palatal expansion (RPE) is a typical orthopedic and orthodontic procedure to correct skeletal transverse maxillary discrepancies and posterior crossbites [1,2,3,4]. RPE appliances generate strong stresses that cause the opening of the midpalatal suture and the increase in maxillary transverse widths [5,6,7]. This stress results in the separation of the two maxillary halves from the midpalatal suture and also expands the nasal cavity. Because of the maxilla’s closeness to the nasal floor, the effects of RPE can influence the morphology of the nasal cavity and therefore the advancement of the nasal cavity’s dimensions [6,8,9,10,11]. The enlargement of the nasal cavity results in an increase in nasal volume and potentially a decrease in nasal resistance and an improvement in the nasal airway [2,6,8,11,12,13,14,15].

For many years, changes in naso-maxillary complex after RPE were assessed through two-dimensional (2D) cephalograms, as well as lateral and posteroanterior cephalometric radiographs, by which it is possible to measure the dimensions of the nasal cavity and airway length [9,10]. However, these images may not be adequate to measure nasal resistance, the size of the airway or the nasal space [9,10,16]. Using three-dimensional (3D) computed tomography (CT), anatomical structures such as facial tissue and the nasal airway may be precisely examined [17]. Additionally, other more accurate techniques, such as rhinomanometry and acoustic rhinometry, were used to access the air pressure and the volume of the nasal cavity [8,13,18,19,20,21]. Recently, the use of 3D CT has led to more accurate results because of its greater resolution, reproducibility and pertinent identification of landmarks [22,23,24]. Cone-beam computed tomography (CBCT), in addition to lower radiation in relation to conventional tomography, enables 3D segmentation and visualization of the nasal airway, as well as the determination of airway volume and surface area [15,25,26,27,28,29,30,31].

Changes in the morphology of the nasal cavity do not always reflect improved respiratory performance. Even though in the literature there are several studies that investigated the relationship between the nasal cavity and respiratory function, the correlation is controversial [16,31,32,33,34]. However, various studies have examined the impact of maxillary expansion on the airway and discovered that the increase in nasal width and volume may result in a decrease in nasal resistance [12,13]. Different studies have found that RME is associated with varying degrees of increased nasal cavity dimensions and decreased nasal obstruction. In recent years, the interest in this topic has increased, and there is still space for more studies to increase the power of the results. The purpose of this prospective clinical trial was to investigate the changes in the dimensions of the naso-maxillary complex in growing patients after RPE using CBCT.

## 2. Methods and Materials

To estimate the sample size, a power analysis was performed using repeated measures ANOVA, and the needed sample size was calculated for an α error prob of 0.05 and a power (1−β) error probability of 95% to include 13 patients. Thus, this prospective clinical study was designed and included 14 subjects (6 boys and 8 girls with a mean age of 10.82 and 11.34 years, respectively), who were treated in the Postgraduate Clinic of the Department of Orthodontics, Aristotle University of Thessaloniki, Greece.

The inclusion criteria were growing patients with unilateral or bilateral crossbite, fully erupted first upper permanent molars who needed RPE as part of their non-extraction orthodontic treatment. Exclusion criteria were non-growing patients or patients with craniofacial anomalies, including cleft lip and palate, multiple missing or dysplastic teeth, patients that had an extraction of at least one permanent tooth, and patients that had a previous orthodontic treatment. The study was approved by the Ethical Committee of the Faculty of Dentistry of the Aristotle University of Thessaloniki in Greece (Protocol nr. 56/05-07-19). An informed consent form was signed by all parents and/or guardians of the patients who participated in the study.

All patients received a Hyrax expander, as the initial part of their treatment, with bands cemented at least on the first maxillary permanent molars and on the first premolars or maxillary deciduous first molars. All Hyrax expanders were manufactured by the same laboratory, and all screws were by the same company. The expansion screw was activated twice a day (0.25 mm per turn, 0.5 mm daily) until the palatal cusp of the maxillary first molar occludes with the buccal cusp of the mandibular first molar. At the end of expansion, the Hyrax screw was stabilized with ligature wire and light-cured composite (Figure 1).

The Hyrax screw remained passive during a 6-month retention period after expansion. Until then, neither fixed nor another orthodontic appliance was used. After the passive period, the Hyrax was removed and replaced by a horse-shoe-type transpalatal arch that was used as a retainer, extending to the palatal surface of the incisors (Figure 2). This type of retainer remains in place until all permanent teeth erupt.

The Hyrax appliance is a tooth borne device for RPE. There are two different tooth born devices, the Haas and Hyrax appliances. The Hyrax device was chosen to be used in this study because this is the most used appliance for RPE since it is more hygienic than Haas for the patient. The Haas device is a fixed split acrylic appliance with acrylic abutting the alveolar ridges, and thus there is a risk of possible inflammation of the palate mucosa.

CBCT scans were obtained before RPE (T1), after the end of expansion (T2) and 6 months post-expansion (T3). The retention period lasted 6 months. That period of retention allowed for reorganization and reossification of the midpalatal suture after expansion [35].

All measurements undertaken on the CBCT images were linear and were made to the nearest 0.01 mm. All radiographic examinations were performed by the same trained technician at the same scanner equipment (Soredex Scanora 3D). The 3D scans were taken in a large field of view (FOV: 75 mm height × 145 mm diameter) in 90 KV, 10 mA and voxel size of 0.35 mm. The CBCT data were exported and analyzed through DICOM viewer OnDemand3d (Cybermed Inc, Daejeon, Republic of Korea).

The measurements of this study were divided into three main parts, such as the dimensions of the nasal width, nasal floor and the aperture of midpalatal suture in different coronal slices of CBCT. All measurements were made by the same examiner on the same day.

The measurements used for assessment of the nasal width were:

1. Nasal Width 1: The maximum lateral interior width of nasal cavity on coronal slice through the center of the mesial buccal root of the first permanent molar (Figure 3).

2. Nasal Width 2: The maximum lateral interior width of nasal cavity on coronal slice through the center of the root of the second premolar (Figure 4).

3. Nasal Width 3: The maximum lateral interior width of nasal cavity on coronal slice through the center of the mesial buccal root of the first premolar (Figure 5).

4. Nasal Width 4: The maximum distance between the inner right surface of nasal cavity and nasal septum on coronal slice through the center of mesial buccal root of the first premolar (Figure 6).

5. Nasal Width 5: The maximum distance between the inner left surface of nasal cavity and nasal septum on coronal slice through the center of mesial buccal root of the first premolar (Figure 7).

6. Nasal Width 6: The maximum lateral interior width of the nasal cavity on coronal slice through the center of the incisive foramen (Figure 8).

The measurements used for assessment of the nasal floor were:

7. Nasal Floor 7: The distance between the junction of anatomical floor of nasal cavity and the medial aspect of the floor ipsilateral sinus on coronal slice through the center of mesial buccal root of the first permanent molar (Figure 9).

8. Nasal Floor 8: The distance between the junction of anatomical floor of nasal cavity and the medial aspect of the floor ipsilateral sinus on coronal slice through the center of root of the second premolar (Figure 10).

9. Nasal Floor 9: The distance between the junction of anatomical floor of nasal cavity and the medial aspect of the floor ipsilateral sinus on coronal slice through the center of root of the central incisors (Figure 11).

The measurements used for assessment of the aperture of midpalatal suture were:

10. Aperture of Midpalatal Suture 10: The distance between the inner points of posterior median palatine suture, bilaterally in posterior nasal spine (PNS) (Figure 12).

11. Aperture of Midpalatal Suture 11: The distance between the inner points of anterior median palatine suture, bilaterally in the inferior alveolar ridge of central incisors (Figure 13).

### 2.1. Statistical Analysis

Evaluation of the mean value variance per measurement at the three time intervals were performed using the paired Wilcoxon signed-rank test. Multiple Pairwise Comparisons (PWC) were performed to assess whether there were any statistically significant differences between the three time intervals. The statistical software SPSS, version 17.0, (SPSS Inc, Chicago III, Chicago, IL, USA) was used for all statistical analyses with *p* < 0.05.

### 2.2. Method of Error

A second measurement was performed on all 14 samples. The second measurements were taken 15 days after the initial ones by the same researcher. There was no statistically significant difference regarding the Intraclass Correlation Coefficient (ICC) that was calculated using the two-way paired t-test between the first and the second measurements. The coincidence of the measurements according to the ICC ranged in very high levels with a very small measurement error (Table 1). All the coefficients for the agreement of the measurements were found to be above 0.96. Conclusively, it can be stated that the method error was very reliable for all measurements.

## 3. Results

Statistically significant differences were found between the mean value of Nasal Widths 1 and 6 between all the time intervals, i.e., immediately after the end of RPE (between T2 and T1), T3 (T3–T1) and 6 months after the end of RPE (T3–T2) (*p* < 0.005), as seen in Table 2 and Table 3. Statistically significant differences were found between the mean value of Nasal Widths 2 and 3 between the time intervals T2–T1 and T3–T1 (*p* < 0.005). On the other hand, no statistically significant decrease was found between the mean value of Nasal Widths 2 and 3 between T3 and T2 (*p* > 0.005), as seen in Table 4 and Table 5. Statistically significant differences were found between the mean value of Nasal Width 4 between the time intervals T2–T1 and T3–T2 (*p* < 0.005). No statistically significant increase was found between the mean value of Nasal Width 4 between T3 and T2 (*p* > 0.005), as seen in Table 6. A statistically significant difference was found only between the mean value of Nasal Width 5 between the time intervals T2 and T1 (*p* < 0.005). No statistically significant differences were found between the mean value of Nasal Width 5 between T3–T1 and T3–T2 (*p* > 0.005), as seen in Table 7.

Statistically significant differences were found between the mean value of Nasal Floor 7 between all the time intervals, i.e., between T2–T1, T3–T1 and T3–T2 (*p* < 0.005). An increase in the mean value of Nasal Floor 7 was found immediately after the end of RPE (between T1 and T2), which was statistically significantly decreased 6 months after RPE (between T2 and T3) (Table 8).

Statistically significant differences were found between the mean value of Nasal Floors 8 and 9 between all the time intervals, i.e., between T2–T1, T3–T1 and T3–T2 (*p* < 0.005). An increase in the mean value of Nasal Floors 8 and 9 was observed immediately after the end of RPE (between T1 and T2), and then they were statistically significantly decreased 6 months after RPE (between T3–T2). The overall (T3–T1) differences for Nasal Floors 8 and 9 were found statistically significantly increased (Table 9 and Table 10).

Statistically significant differences were found between the mean value of Apertures of Midpalatal Sutures 10 and 11 between all the time intervals, i.e., between T2–T1, T3–T1 and T3–T2 (*p* < 0.005). An increase in the mean values of Apertures of Midpalatal Sutures 10 and 11 were observed immediately after the end of RPE (T2–T1). Statistically significant decreases were found 6 months after RPE (T3–T2). The overall (T3–T1) differences between pre-expansion and 6 months after expansion for Apertures of Midpalatal Sutures 10 and 11 were found statistically significantly increased (Table 11 and Table 12).

The calculated mean values for each measurement in the three time intervals, as well as the differences between T2–T1, T3–T1 and T3–T2, are shown in Table 13.

A statistically significant increase in all measurements of the nasal width, nasal floor and the aperture of the midpalatal suture is observed immediately after RPE expansion (T2–T1). An increase in all measurements is also observed six months after expansion (T3–T1), but with no statistical significance of Nasal Widths 4 and 5. A decrease in all measurements is seen between the end of expansion and 6 months later.

## 4. Discussion

The current prospective clinical study aimed to evaluate transverse maxilla and nasal changes in growing patients treated with a Hyrax device using images obtained by Cone-Beam Computed Tomography (CBCT).

The results of this study show that the expansion of the maxilla, shown at the aperture of the midpalatal suture, has a triangular pattern. It was found that the amount of increase was greater (a) at the anterior median palatine suture, (b) bilaterally in the inferior alveolar ridge of the central incisors than the posterior, and (c) bilaterally in at posterior nasal spine (PNS). Similar results to our study have shown that the midpalatal suture has a triangle shape, with the triangle’s vertex in the PNS and its base in perspective [1,2,36,37]. According to Da Silva et al. [38], the palatal suture has its largest width in the anterior region and does not have a parallel opening configuration in the axial plane. Additionally, Kartalian et al. [39] found that after RPE, the midpalatal suture separates in a nonparallel manner in response to expansion forces. The maxilla articulates with unpaired bones, which limits the amount of separation. Caldas et al. [40], studying the effect of rapid maxillary expansion on the nasal cavity assessed with cone-beam computed tomography, concluded that there is a great transverse movement and subsequent separation of the nasal conchae from the nasal septum in both the anterior and posterior sections of the inferior portion of the lateral walls of the nasal cavity after RPE. 

In the literature, there is disagreement regarding this triangle-shaped arrangement because some studies have indicated that the suture edges expand up parallel to each other [41,42,43]. According to Zeng et al. [44], the expansion of the midpalatal suture may follow the parallel opening pattern. This may have been influenced by the location of the screws and the patient’s age. However, according to two systematic reviews from the last decade, there is inconsistent evidence for either a parallel or triangular arrangement of the median palatine suture opening [45,46].

In the current study, a statistically significant increase was found in all measurements of the nasal width, nasal floor and the aperture of the midpalatal suture immediately after RPE expansion (T2–T1) (Table 2). Previous studies evaluating nasal changes after RPE showed similar results to our study [4,35,40,44,47,48]. Fastuca et al. [48] found that there was no statistically significant difference between the nasal size increments obtained using the Hyrax-type appliance and the Haas-type appliance both anchored to deciduous teeth. 

During the retention time between the end of RPE and 6 months post-expansion (T3–T2), a decrease in all variables was found. According to Ballanti et al. [35], after a 6-month retention period, the midpalatal suture lost 89.55 and 77.78% of its increased area at the anterior nasal spine (ANS) and the posterior nasal spine (PNS), respectively. They believed that after the 6-month retention period, the midpalatal suture looked restructured with a transverse dimension equivalent to the pre-treatment width. In the present study, after 6 months retention the midpalatal suture lost 54.485% at ANS and 36.842% at PNS, respectively.

Evaluating the changes between the start of the treatment and the 6-month post-expansion (T3–T1), all variables showed a considerable increase, which demonstrates a significant amount of expansion of the nasal cavity. The expansion of the nasal cavity resulted in stability at the end of the 6-month retention period. This is in accordance with the results of Ballanti et al. [35]. In their study, they found that the increase in the transverse dimensions of the nasal cavity immediately after RPE remained stable for 6 months of retention. In a similar study by Palaisa et al. [49], who used conventional tomography (CT) to evaluate the changes in the nasal cavity after RPE, it was found that the expansion of the nasal cavity was stable after a 3-month retention period. Truong et al. [50] found that the increase in the nasal cavity remains stable approximately two and a half years after the end of RPE.

A regularly seen observation in the literature suggests that the structural enlargement of the nasal cavity may be the cause of the decrease in nasal airway resistance [6,18]. The same results were concluded by El et al. (2014) [31], who found that RPE may be able to enhance breathing by lowering nasal resistance. There is a correlation between the significant gain in the inner nasal cavity dimensions after RPE and an improvement in mouth breathers’ breathing patterns [15]. In the same way, the existence of transverse disorders of the maxilla (such as crossbite) is a risk factor for abnormal breathing. Since the maxilla and the nasal airway are directly connected anatomically, one would anticipate that these problems may increase upper airway resistance and may contribute to the development of obstructive sleep apnea [51].

Obstructive sleep apnea (OSA) is a sleep-related breathing disorder, a chronic disease characterized by repetitive episodes of sleep-related breathing pauses. OSA affects the physical and mental well-being of patients, negatively affecting their quality of life. Recurrent episodes of OSA may consequently lead to daytime sleepiness, fatigue, growth retardation, mood and behavioral disturbances, hyperactivity, learning and memory deficits, increased risk taking and injury risk, and poor neurocognitive performance. The impact of OSA on quality of life may have further long-term implications such as endocrine and cardiovascular disorders [52,53,54,55,56].

Overall, this study is clinically relevant in the field, considering the impacts of maxillary transversal width on breathing patterns and obstructive sleep apnea. These matters directly impact children and adolescents’ health and quality of life; therefore, studies on this topic are critical to filling still-existing gaps in the literature.

It is important to note that although the gain in the inner dimensions of the nasal cavity following RPE may be significant, its usage for respiratory purposes exclusively is not justified, according to the literature. Recent research indicated an improvement in mouth breathers’ breathing patterns [26,31]. The lack of scientific proof that growing patients treated with RPE maintain respiratory gains after a follow-up period emphasizes the necessity for additional long-term investigations.

The limitation of this study is the sample size. Although the power analysis detected that the number of subjects was sufficient to conduct this study, future studies with a larger number of patients will restrict this limitation.

## 5. Conclusions

The results of this study suggest that RPE produces a significant amount of expansion of the nasal cavity (nasal width, nasal floor) and aperture of the midpalatal suture in growing patients. The skeletal expansion of the maxilla was found to have a triangular pattern.

## Figures and Tables

**Figure 1 diagnostics-13-01322-f001:**
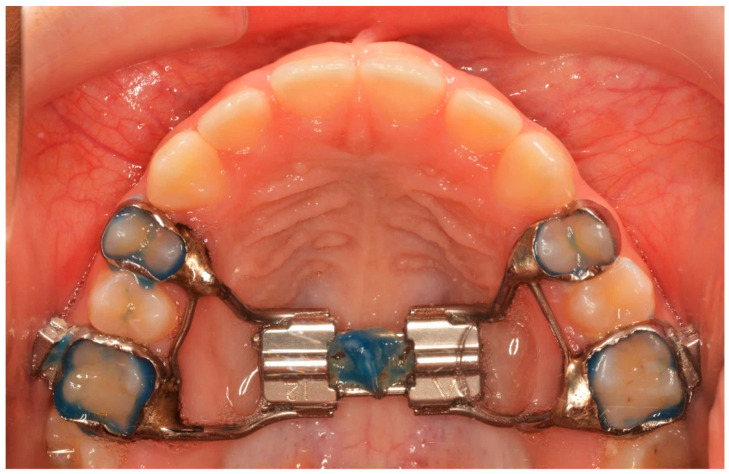
The end of the active phase of expansion.

**Figure 2 diagnostics-13-01322-f002:**
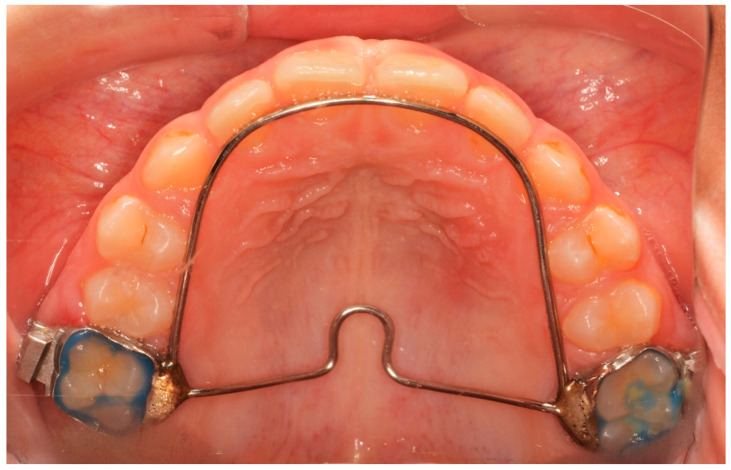
Horse-shoe-type TPA used after T2 time interval.

**Figure 3 diagnostics-13-01322-f003:**
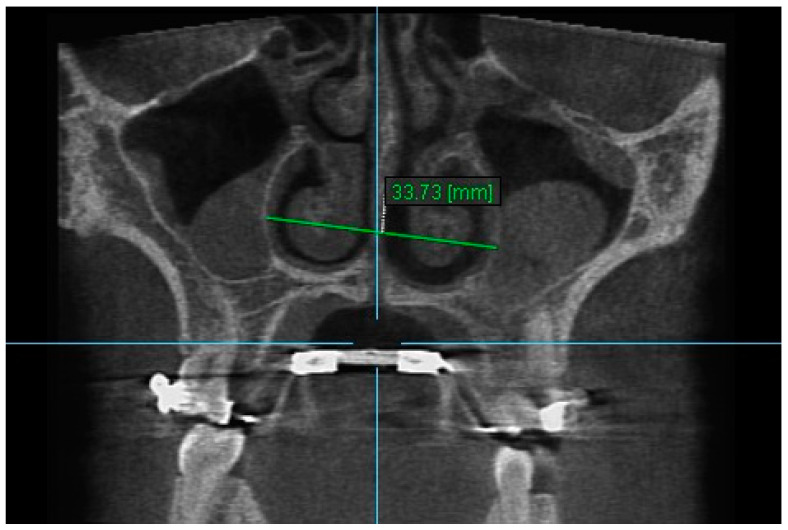
Nasal Width 1.

**Figure 4 diagnostics-13-01322-f004:**
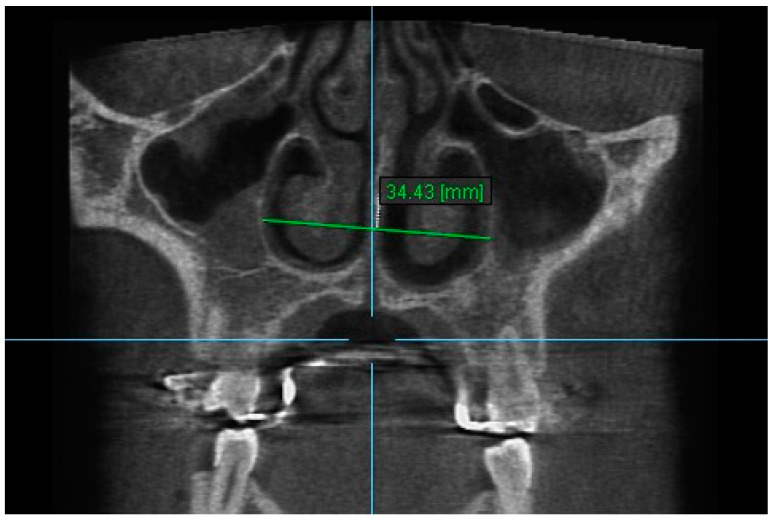
Nasal Width 2.

**Figure 5 diagnostics-13-01322-f005:**
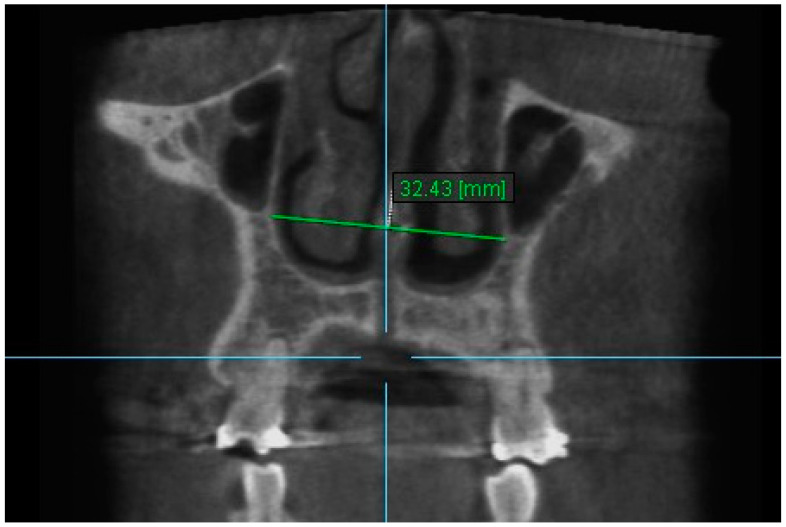
Nasal Width 3.

**Figure 6 diagnostics-13-01322-f006:**
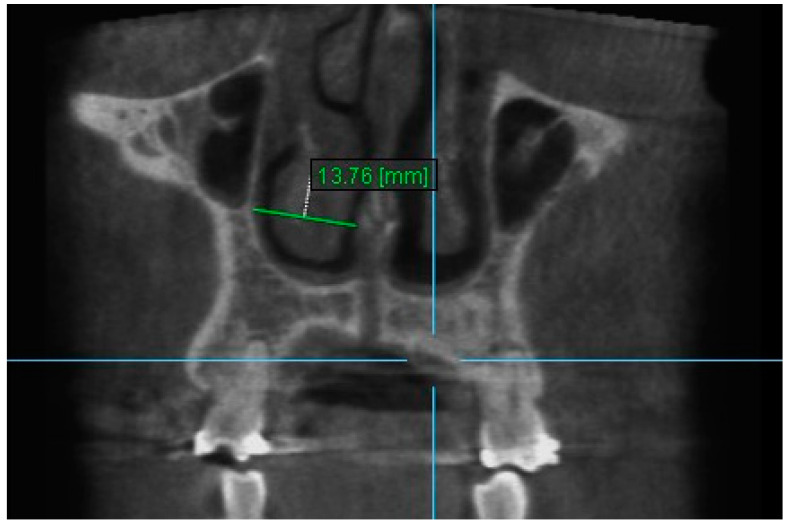
Nasal Width 4.

**Figure 7 diagnostics-13-01322-f007:**
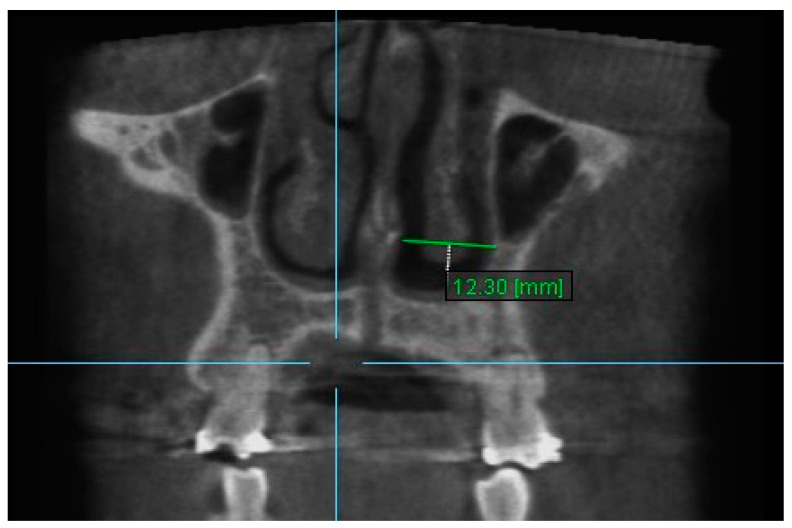
Nasal Width 5.

**Figure 8 diagnostics-13-01322-f008:**
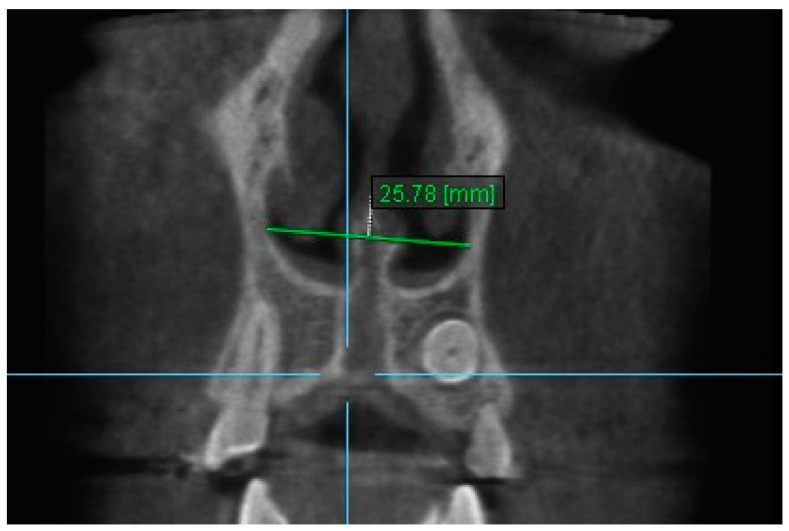
Nasal Width 6.

**Figure 9 diagnostics-13-01322-f009:**
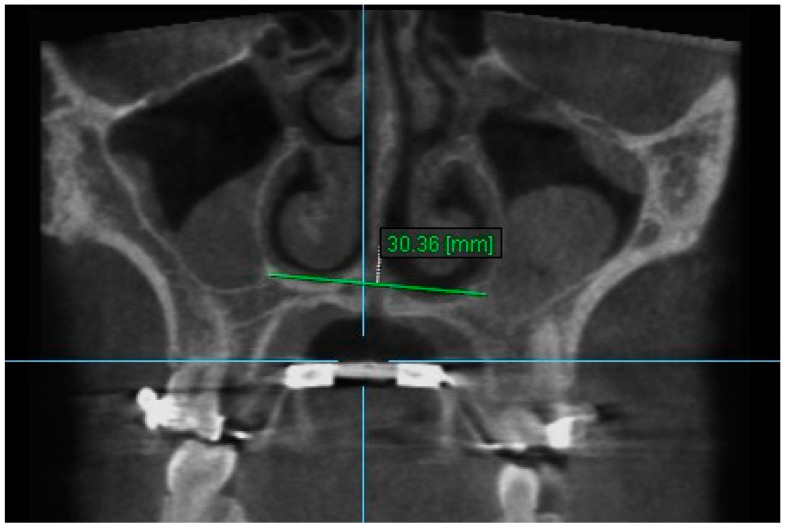
Nasal Floor 7.

**Figure 10 diagnostics-13-01322-f010:**
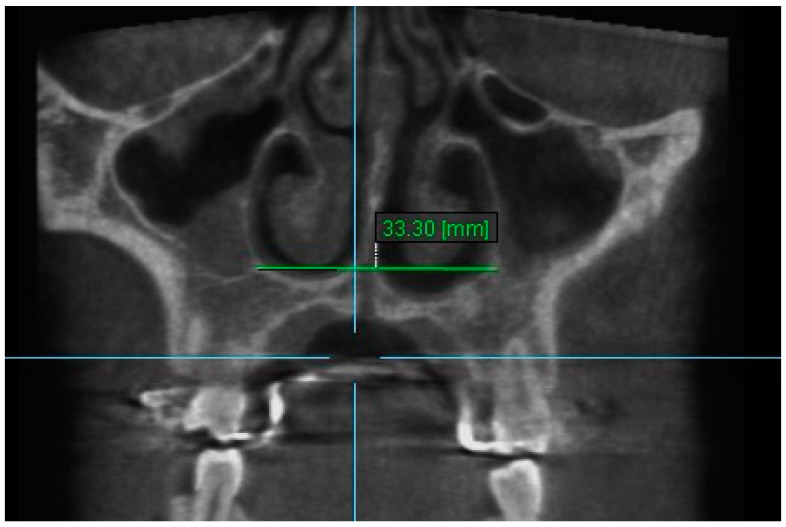
Nasal Floor 8.

**Figure 11 diagnostics-13-01322-f011:**
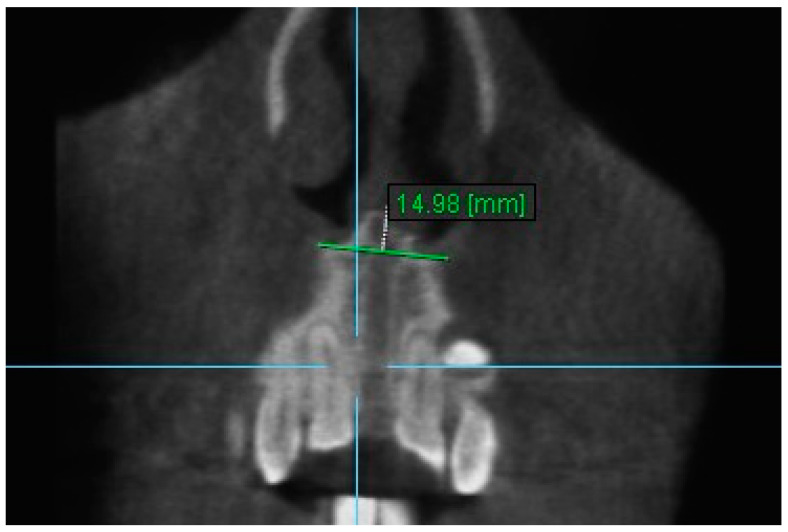
Nasal Floor 9.

**Figure 12 diagnostics-13-01322-f012:**
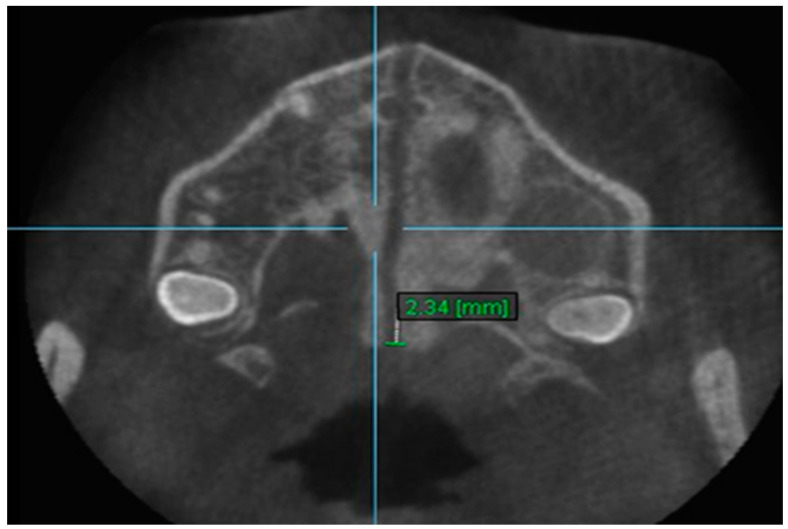
Aperture of Midpalatal Suture 10.

**Figure 13 diagnostics-13-01322-f013:**
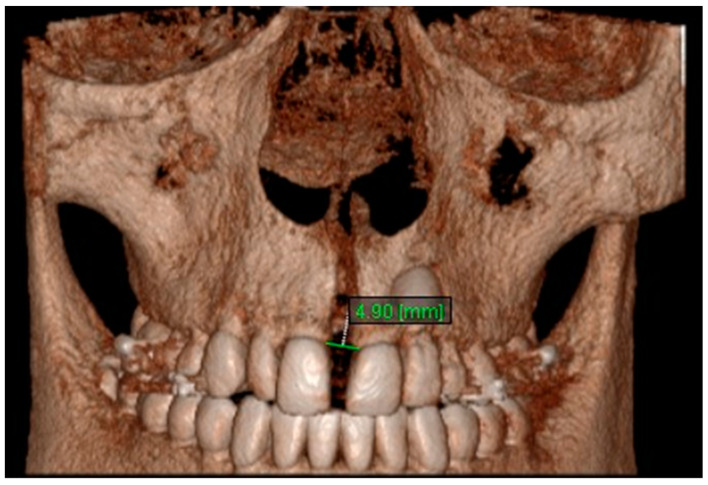
Aperture of Midpalatal Suture 11.

**Table 1 diagnostics-13-01322-t001:** Statistical indicators for the method of error for every parameter and time Interval.

Variables	Time Interval	1st Measurement (mm)	2nd Measurement (mm)	ICC (95% CI) *
Nasal Width 1	T1	30.36 ± 2.76	30.44 ± 2.69	0.998 (0.992–0.999)
	T2	32.46± 2.69	32.47 ± 2.57	0.997 (0.989–0.999)
	T3	31.71 ± 2.76	31.72 ± 2.77	0.998 (0.993–0.999)
Nasal Width 2	T1	30.54 ± 3.09	30.60 ± 3.03	0.996 (0.990–0.999)
	T2	32.59 ± 3.10	32.69 ± 3.13	0.997 (0.992–0.999)
	T3	31.73 ± 2.88	31.71 ± 2.90	0.994 (0.981–0.998)
Nasal Width 3	T1	28.32 ± 3.13	28.34 ± 2.98	0.996 (0.988–0.998)
	T2	30.83 ± 2.91	35.93 ± 2.90	0.993 (0.980–0.998)
	T3	29.69 ± 2.65	29.65 ± 2.65	0.990 (0.968–0.996)
Nasal width 4	T1	12.84 ± 1.58	12.83 ± 1.51	0.982 (0.943–0.994)
	T2	13.75 ± 1.33	13.59 ± 1.25	0.988 (0.947–0.996)
	T3	13.17 ± 1.66	13.21 ± 1.70	0.988 (0.962–0.996)
Nasal Width 5	T1	12.84 ± 1.84	12.83 ± 1.71	0.989 (0.967–0.996)
	T2	14.21 ± 1.87	13.99 ± 1.85	0.982 (0.915–0.995)
	T3	13.28 ± 1.34	12.90 ± 1.12	0.993 (0.976–0.998)
Nasal Width 6	T1	23.58 ± 1.52	23.70 ± 1.51	0.979 (0.934–0.993)
	T2	26.61 ± 1.79	26.65 ± 1.69	0.991 (0.970–0.997)
	T3	25.16 ± 1.70	25.25 ± 1.76	0.988 (0.965–0.996)
Nasal Floor 7	T1	28.62 ± 2.81	28.68 ± 2.83	0.998 (0.994–0.999)
	T2	30.84 ± 2.57	30.89 ± 2.59	0.997 (0.993–0.999)
	T3	30.28 ± 2.86	30.19 ± 2.76	0.995 (0.985–0.998)
Nasal Floor 8	T1	28.44 ± 3.95	28.39 ± 3.90	0.997 (0.992–0.999)
	T2	30.92 ± 4.23	30.82 ± 4.18	0.997 (0.993–0.999)
	T3	29.99 ± 4.21	29.99 ± 4.11	0.995 (0.987–0.998)
Nasal Floor 9	T1	15.75 ± 3.69	15.67 ± 3.51	0.995 (0.984–0.998)
	T2	18.52 ± 3.21	18.42 ± 2.88	0.991 (0.974–0.997)
	T3	17.18 ± 3.51	17.37 ± 3.39	0.990 (0.958–0.997)
Aperture of Midpalatal Suture 10	T1	0.00 ± 0.00	0.00 ± 0.00	0
	T2	2.52 ± 0.92	2.54 ± 0.98	0.993 (0.978–0.997)
	T3	1.65 ± 0.77	1.67 ± 0.76	0.982 (0.945–0.994)
Aperture of Midpalatal Suture 11	T1	0.00 ± 0.00	0.00 ± 0.00	0
	T2	5.46 ± 1.51	5.24 ± 1.41	0.973 (0.920–0.994)
	T3	2.48 ± 0.78	2.40 ± 0.79	0.994 (0.982–0.998)

T1: before RPE, T2: after the end of expansion T3: 6 months post-expansion; ICC: Intraclass Correlation Coefficient, 95% CI = 95% Confidence Interval. * statistically significant.

**Table 2 diagnostics-13-01322-t002:** Results of the mean differences (mm) of Nasal Width 1 in the three time intervals.

Time Interval	Mean Difference	Standard Deviation	95% Confidence Interval	*p* Value	*p* Significance
T2–T1	2.13	0.82	2.07–2.19	<0.001	***
T3–T1	1.30	1.02	1.26–1.34	<0.001	***
T3–T2	−0.83	0.59	−0.85–−0.81	0.003	**

NS: not statistically significant. **, *** level of statistical significance: *** when *p* < 0.001 there is very strong evidence, ** when 0.001 ≤ *p* < 0.01 there is strong evidence.

**Table 3 diagnostics-13-01322-t003:** Results of the mean differences (mm) of Nasal Width 6 in the three time intervals.

Time Interval	Mean Difference	Standard Deviation	95% Confidence Interval	*p* Value	*p* Significance
T2–T1	2.92	1.46	2.74–3.09	<0.001	***
T3–T1	1.52	0.99	1.39–1.64	0.003	**
T3–T2	−1.39	1.38	−1.45–−1.35	0.003	**

NS: not statistically significant. **, *** level of statistical significance: *** when *p* < 0.001 there is very strong evidence, ** when 0.001 ≤ *p* < 0.01 there is strong evidence.

**Table 4 diagnostics-13-01322-t004:** Results of the mean differences (mm) of Nasal Width 2 in the three time intervals.

Time Interval	Mean Difference	Standard Deviation	95% Confidence Interval	*p* Value	*p* Significance
T2–T1	2.08	0.69	2.05–2.10	<0.001	***
T3–T1	1.28	1.15	1.61–1.41	<0.001	***
T3–T2	−0.79	0.093	−0.94–−0.64	0.074	NS

NS: not statistically significant. *** level of statistical significance: *** when *p* < 0.001 there is very strong evidence.

**Table 5 diagnostics-13-01322-t005:** Results of the mean differences (mm) of Nasal Width 3 in the three time intervals.

Time Interval	Mean Difference	Standard Deviation	95% Confidence Interval	*p* Value	*p* Significance
T2–T1	2.77	1.65	2.68–2.86	<0.001	***
T3–T1	1.61	1.16	1.35–1.88	<0.001	***
T3–T2	−1.15	1.67	−1.34–−0.94	0.061	NS

NS: not statistically significant. *** level of statistical significance: *** when *p* < 0.001 there is very strong evidence.

**Table 6 diagnostics-13-01322-t006:** Results of the mean differences (mm) of Nasal Width 4 in the three time intervals.

Time Interval	Mean Difference	Standard Deviation	95% Confidence Interval	*p* Value	*p* Significance
T2–T1	1.05	1.01	0.91–1.19	0.012	*
T3–T1	0.28	0.77	0.24–0.33	0.651	NS
T3–T2	−0.76	0.88	−0.95–−0.58	0.036	*

NS: not statistically significant. * level of statistical significance: * when 0.01 ≤ *p* < 0.05 there is moderate evidence.

**Table 7 diagnostics-13-01322-t007:** Results of the mean differences (mm) of Nasal Width 5 in the three time intervals.

Time Interval	Mean Difference	Standard Deviation	95% Confidence Interval	*p* Value	*p* Significance
T2–T1	1.39	0.70	1.37–1.40	0.004	**
T3–T1	0.68	0.88	0.41–0.95	0.077	NS
T3–T2	−0.71	1.18	−1.01–−0.42	0.327	NS

NS: not statistically significant. ** level of statistical significance: ** when 0.001 ≤ *p* < 0.01 there is strong evidence.

**Table 8 diagnostics-13-01322-t008:** Results of the mean differences (mm) of Nasal Floor 7 in the three time intervals.

Time Interval	Mean Difference	Standard Deviation	95% Confidence Interval	*p* Value	*p* Significance
T2–T1	2.32	1.03	2.18–2.47	<0.001	***
T3–T1	1.59	1.26	1.57–1.61	<0.002	**
T3–T2	−0.73	0.62	−0.89–−0.57	0.004	**

NS: not statistically significant. **, *** level of statistical significance: *** when *p* < 0.001 there is very strong evidence, ** when 0.001 ≤ *p* < 0.01 there is strong evidence.

**Table 9 diagnostics-13-01322-t009:** Results of the mean differences (mm) of Nasal Floor 8 in the three time intervals.

Time Interval	Mean Difference	Standard Deviation	95% Confidence Interval	*p* Value	*p* Significance
T2–T1	2.37	0.91	2.19–2.55	<0.001	***
T3–T1	1.45	0.89	1.28–1.61	0.003	***
T3–T2	−0.92	0.62	−0.94–−0.91	0.001	**

NS: not statistically significant. **, *** level of statistical significance: *** when *p* < 0.001 there is very strong evidence, ** when 0.001 ≤ *p* < 0.01 there is strong evidence.

**Table 10 diagnostics-13-01322-t010:** Results of the mean differences (mm) of Nasal Floor 9 in the three time intervals.

Time Interval	Mean Difference	Standard Deviation	95% Confidence Interval	*p* Value	*p* Significance
T2–T1	2.99	1.48	2.72–3.27	0.005	**
T3–T1	1.51	0.97	1.40–1.61	<0.002	**
T3–T2	−1.49	1.54	−1.66–−1.32	0.001	**

NS: not statistically significant. ** level of statistical significance: ** when 0.001 ≤ *p* < 0.01 there is strong evidence.

**Table 11 diagnostics-13-01322-t011:** Results of the mean differences (mm) of Aperture of Midpalatal Suture 10 in the three time intervals.

Time Interval	Mean Difference	Standard Deviation	95% Confidence Interval	*p* Value	*p* Significance
T2–T1	1.90	0.88	1.37–2.43	0.003	**
T3–T1	1.16	0.72	0.72–1.59	0.003	**
T3–T2	−0.75	0.49	−0.84–−0.65	0.003	**

NS: not statistically significant. ** level of statistical significance: ** when 0.001 ≤ *p* < 0.01 there is strong evidence.

**Table 12 diagnostics-13-01322-t012:** Results of the mean differences (mm) of Aperture of Midpalatal Suture 11.

Time Interval	Mean Difference	Standard Deviation	95% Confidence Interval	*p* Value	*p* Significance
T2–T1	4.57	1.41	3.73–5.41	0.003	**
T3–T1	2.08	0.76	1.62–2.53	0.003	**
T3–T2	−2.49	1.07	−2.88–−2.11	0.003	**

NS: not statistically significant. ** level of statistical significance: ** when 0.001 ≤ *p* < 0.01 there is strong evidence.

**Table 13 diagnostics-13-01322-t013:** Total outcomes (mm) of the mean values of all the variables and their differences in the three time intervals.

Variables	Mean T1	Mean T2	Mean T3	Mean Difference (T2-T1)	*p*	Mean Difference (T3-T1)	*p*	Mean Difference (T3-T2)	*p*
Nasal Width 1	28.83	30.97	30.14	2.13	***	1.30	***	−0.83	**
Nasal Width 2	28.84	30.92	30.13	2.08	***	1.28	***	−0.79	(NS)
Nasal Width 3	26.45	29.22	28.07	2.77	***	1.61	***	−1.15	(NS)
Nasal Width 4	11.92	12.97	12.21	1.05	*	0.28	(NS)	−0.76	*
Nasal Width 5	11.75	13.14	12.43	1.39	*	0.68	(NS)	−0.71	(NS)
Nasal Width 6	22.73	25.64	24.24	2.92	***	1.52	**	−1.39	**
Nasal Floor 7	27.05	29.37	28.61	2.32	***	1.59	***	−0.73	**
Nasal Floor 8	26.27	28.64	27.72	2.37	***	1.45	**	−0.92	**
Nasal Floor 9	13.61	16.60	15.11	2.99	**	1.51	***	−1.49	**
Aperture of Midpalatal Suture 10	0	1.90	1.16	1.90	**	1.16	**	−0.75	**
Aperture of Midpalatal Suture 11	0	4.57	2.08	4.57	***	2.08	**	−2.49	***

NS: not statistically significant. *, **, *** level of statistical significance. *** when *p* < 0.001 there is very strong evidence, ** when 0.001 ≤ *p* < 0.01 there is strong evidence, * when 0.01 ≤ *p* < 0.05 there is moderate evidence.

## Data Availability

The data presented in this study are available on request from the corresponding author. The data are not publicly available due to privacy reasons.

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
