# Peer review of "Three-Dimensional Cone-Beam Computed Tomography Evaluation of Changes in Naso-Maxillary Complex Associated with Rapid Palatal Expansion"

_diagnostics, 2023, doi:10.3390/diagnostics13071322_

Round 1
Reviewer 1 Report
I was pleased to review the manuscript “diagnostics-2291533” entitled “Three-dimensional cone beam computed tomography evaluation of changes in naso-maxillary complex associated with rapid palatal expansion” for the Diagnostics journal. This clinical trial assessed naso-maxillary dimensions (nasal width, nasal floor and aperture of midpalatal suture) changes in growing patients after rapid palatal expansion evaluated by Cone Beam Computed Tomography. Overall, this study is clinically relevant in the field, considering the impacts of maxillary transversal width on breathing patterns and obstructive sleep apnea. These matters directly impact children and adolescents’ health and quality of life; therefore, studies on that topic are critical to filling still-existing gaps in the literature. However, I suggest a major revision to facilitate and improve the reading and clarify the importance of the new findings compared to those that have already been published.
Please refer to the comments below:
- Introduction:
What gap in the literature does the present study address? Please write about it: what’s the difference between this study and the others already published?
What is the purpose of this study?
Please provide more information about this study’s originality and relevance to the field.
- Methods and Materials:
How long did the patients use the horse-shoe type transpalatal arch?
Please provide a footnote in table 1 with the meanings of T1, T2, T3, and ICC.
- Results:
Please provide a footnote in table 2 with the meanings of T1, T2, T3, and the symbols (*).
Please provide the p values in the text when statistically significant differences are mentioned (lines 176-180).
- Discussion:
The phrase “Caldas et al. [40] in their study aimed to evaluate the effects of RPE with Haas expander on linear dimensions of the nasal cavity using CBCT concluded that in both anterior and posterior sections of the inferior portion of the lateral walls of the nasal cavity there is a great transverse movement and subsequent separation of the nasal conchae from the nasal septum.” in lines 194-198 is confusing. Please rephrase it to be more straightforward.
The authors mention that the previous study shows no difference between nasal size increments when using Hyrax or Haas appliances. What about the other measurements? Does the device have an impact on the midpalatal suture aperture? Why did the authors choose to use the Hyrax appliance? What are the advantages and disadvantages of Hyrax and Haas? Please discuss these points in the manuscript.
How much percent did the midpalatal suture lose after the 6-month retention in the present study (T3-T2)? Is it similar to the study of Ballanti et al.? What’s the benefit of the treatment with RPE for crossbite if, after the retention, the transverse dimension in the maxillary arch is equivalent to the one before the treatment?
What are the impacts of the present results on patients’ health and quality of life? Explain the effects of obstructive sleep apnea and abnormal breathing in the life of growing patients.
Please write a paragraph with the limitations of the study.
Author Response
Dear reviewer,
Thank very much for taking the time to review the manuscript “diagnostics-2291533” entitled “Three-dimensional cone beam computed tomography evaluation of changes in naso-maxillary complex associated with rapid palatal expansion” for the Diagnostics journal.
Also, thank you for your valuable comments and suggestions.
Below please find my response with bold letters.
Suggestions for Authors
I was pleased to review the manuscript “diagnostics-2291533” entitled “Three-dimensional cone beam computed tomography evaluation of changes in naso-maxillary complex associated with rapid palatal expansion” for the Diagnostics journal. This clinical trial assessed naso-maxillary dimensions (nasal width, nasal floor and aperture of midpalatal suture) changes in growing patients after rapid palatal expansion evaluated by Cone Beam Computed Tomography. Overall, this study is clinically relevant in the field, considering the impacts of maxillary transversal width on breathing patterns and obstructive sleep apnea. These matters directly impact children and adolescents’ health and quality of life; therefore, studies on that topic are critical to filling still-existing gaps in the literature. However, I suggest a major revision to facilitate and improve the reading and clarify the importance of the new findings compared to those that have already been published.
Please refer to the comments below:
- Introduction:
What gap in the literature does the present study address? Please write about it: what’s the difference between this study and the others already published?
The difference between this study and the others is that for many years the evaluation of changes in naso-maxillary complex after RPE became were assessed through two-dimensional (2D) cephalograms, lateral and posteroanterior cephalometric radiographs, which is possible to measure the dimensions of nasal cavity and airway length. Using three-dimensional (3D) computed tomography (CT), anatomical structures such as face tissue and the nasal airway may be precisely examined. Recently, the use of 3D CT three-dimensional computed tomography leads to more accurate results be-cause of its greater resolution, reproducibility and pertinent identification of land-marks [22-24]. Cone beam computed tomography (CBCT) technology, except the lower radiation in relation to conventional tomography, enables 3D segmentation and visualization of the nasal airway, as well as the determination of airway volume and surface area. Lines: 42-55.
Even though in the literature there are several studies investigated the relationship between nasal cavity and respiratory function, the correlation is controversial [16,31-34. However, various studies have examined the impact of maxillary expansion on the airway and discovered that the increase advance of the nasal width and volume may have as a result to a the decrease of nasal the resistance [12,13]. Different studies conducted that the RME is associated to varying degrees of increased nasal cavity dimension and decreased nasal obstruction. Even if the last years the interest of this top-ic has increased, there is still space for more studies to increase the power of the re-sults. Lines: 58-65.
What is the purpose of this study?
The purpose of this prospective clinical trial was to investigate the changes in the dimensions of naso-maxillary complex in growing patients after RPE using CBCT. This is written in the abstract of the study but I also, added in the end of the introduction.
Please provide more information about this study’s originality and relevance to the field.
I provide more information in the introduction section.
- Methods and Materials:
How long did the patients use the horse-shoe type transpalatal arch?
The horse-shoe type transpalatal arch remain in place until all permanent teeth erupt. Line 97-98.
Please provide a footnote in table 1 with the meanings of T1, T2, T3, and ICC.
I did so.
- Results:
Please provide a footnote in table 2 with the meanings of T1, T2, T3, and the symbols (*).
I did so.
Please provide the p values in the text when statistically significant differences are mentioned (lines 176-180).
To provide the p values of the statistically significant differences measurements mention between lines 176-180 I have provided in the text the tables 2-12. Lines 212-252.
- Discussion:
The phrase “Caldas et al. [40] in their study aimed to evaluate the effects of RPE with Haas expander on linear dimensions of the nasal cavity using CBCT concluded that in both anterior and posterior sections of the inferior portion of the lateral walls of the nasal cavity there is a great transverse movement and subsequent separation of the nasal conchae from the nasal septum.” in lines 194-198 is confusing. Please rephrase it to be more straightforward.
I rephrase as advised. Lines 280-285.
The authors mention that the previous study shows no difference between nasal size increments when using Hyrax or Haas appliances. What about the other measurements? Does the device have an impact on the midpalatal suture aperture? Why did the authors choose to use the Hyrax appliance? What are the advantages and disadvantages of Hyrax and Haas? Please discuss these points in the manuscript.
Yes, the device has an impact on the midpalatal suture aperture which is shown at tables 11,12 an 13. These tables are added to the revise manuscript. Also, the impact of RPE on the midpalatal suture aperture has been discuss in different places in the discussion section. For example: “The results of this study show that the expansion of the maxilla, shown at aper-ture of mid-palatal suture, has a triangular pattern. It was found that the amount of increase was greater at (a)the anterior median palatine suture, (b)bilaterally in the inferior alveolar ridge of central incisors than the posterior, and (c) bilaterally in posterior nasal spine (PNS).” Lines: 270-274.
I add the lines 101-105 in the method and material section to discuss the advantages and disadvantages of Hyrax and Haas appliances.
How much percent did the midpalatal suture lose after the 6-month retention in the present study (T3-T2)? Is it similar to the study of Ballanti et al.? What’s the benefit of the treatment with RPE for crossbite if, after the retention, the transverse dimension in the maxillary arch is equivalent to the one before the treatment?
After 6 months retention midpalatal suture at PNS lost 0,75mm from 1,90 mm that gain from RPE (Table 11) mean 36,842% and at ANS lost 2,49 from 4,57 (Table12) mean 54,485%. I added this information to the discussion section.
The aim of Ballanti et al. study was to use low-dose coronal computed tomography (CT) scans to evaluate the treatment and postretention effects of rapid maxillary expansion (RME) on the maxillary central incisors, midpalatal suture, and nasal cavity. Multi-slice coronal CT scans of 17 subjects (7 boys, 10 girls; mean age, 11.2 years) were taken before RME (T0), at the end of active expansion phase (T1), and after the retention period of 6 months (T2). They concluded that the maxillary halves were separated by RME in a parallel manner. At T2 after RME therapy, the suture appeared reorganized, and the expansion of the nasal cavity was stable. Our study aimed to evaluate the effect of RPE on the naso-maxillary complex, meaning nasal cavity (nasal width and nasal floor) and the midpalatal suture (aperture of midpalatal suture). Thus, is similar to study of the Ballanti et al.
In our study we are not evaluating changes of the maxillary arch after PRE. We are studying the changes of the nasal cavity and midpalatal suture. We have another study, using the same sample, evaluating the sort-term effects of RPE on the maxillary arch transverse dimensions.
What are the impacts of the present results on patients’ health and quality of life? Explain the effects of obstructive sleep apnea and abnormal breathing in the life of growing patients.
To explain the effects of obstructive sleep apnea in the life of growing patients The following paragraph is added. Lines 331-338.
Obstructive sleep apnoea (OSA) is a sleep-related breathing disorder, chronic dis-ease characterized by repetitive episodes of sleep-related breathing pauses. OSA affects physical and mental well-being of patients negatively affect their quality of life. Recurrent episodes of OSA may consequently lead to daytime sleepiness, fatigue, growth retardation, mood and behavioral disturbances, hyperactivity, learning and memory deficits and increased risk taking and injury risk and poor neurocognitive performance. The impact of OSA on quality of life may have further long-term impli-cations like endocrine and cardiovascular disorders [52-56].
Also, I add the following references:
- Oniani, N., Sakhelashvili, I., Supatashvili, M., Basishvili, T., Eliozishvili, M., Maisuradze, L., & Cervena, K. Relationship between sleep disorders and health related quality of life-results from the Georgia SOMNUS study. International Journal of Environmental Research and Public Health, 2018, 15(8), 1588. https://doi.org/10.3390/ijerph15081588.
- Vogler K., Daboul A., Obst A., Fietze I., Ewert R., Biffar R., Krüger M. Quality of life in patients with obstructive sleep apnea: Results from the study of health in Pomerania. Journal of Sleep Research 2023, 32 (1) https://doi.org/10.1111/jsr.13702
- Lynch M., Lindsey C. Elliott, BA, Avis KT, Schwebel D.C., Goodin B.R. Quality of life in youth with obstructive sleep apnea syndrome (OSAS) treated with continuous positive airway pressure (CPAP) therapy. Behav Sleep Med. 2019 ; 17(3): 238–245. doi:10.1080/15402002.2017.1326918.Franco RA, Rosenfeld RM, & Rao M. Quality of life for children with obstructive sleep apnea. Otolaryngol. Head Neck Surg, 2000, 123, 9–16. [PubMed: 10889473]
- Marcus CL, Carroll JL, Koerner CB, Hamer A, Lutz J, & Loughlin GM. Determinants of growth in children with the obstructive sleep apnea syndrome. The Journal of Pediatrics, 1994, 125 (4), 556–562. [PubMed: 7931873]
- Melendres CS, Lutz JM, Rubin ED, & Marcus CL. Daytime sleepiness and hyperactivity in children with suspected sleep-disordered breathing. Pediatrics, 2004, 114, 768–775. [PubMed:15342852]
Please write a paragraph with the limitations of the study.
Lines 345-347 were added in the discussion to describe the limitations of the study.
Reviewer 2 Report
Thank you for the opportunity to review this paper.
The article does not present an exhaustive introduction. Materials and methods should be improved and accurately describe the procedures and characteristics of the radiological equipment and the FOV used.
In consideration of the previous literature it would be more appropriate to carry out a reorientation of the Dicom files from the Cone-Beam at T0 and a Voxel Based Superimposition with the subsequent timing, before measurements.
It is recommended to reorient the dicom files of the skull with previously validated protocols:
-PMID: 35691961 PMCID: PMC9189077 DOI: 10.1186/S40510-022-00413-8
- PMID: 23623785 DOI: 10.1016/J.IJOM.2013.03,007
Scientific English in the text should be improved.
Author Response
Thank very much for taking the time to review the manuscript “diagnostics-2291533” entitled “Three-dimensional cone beam computed tomography evaluation of changes in naso-maxillary complex associated with rapid palatal expansion” for the Diagnostics journal.
Also, thank you for your valuable comments and suggestions.
Below please find my response with bold letters.
The article does not present an exhaustive introduction. Materials and methods should be improved and accurately describe the procedures and characteristics of the radiological equipment and the FOV used.
I improve the introduction.
In material and method section I describe the procedures and characteristics of the radiological equipment and the FOV used as asked. Lines: 107-112. All measurements undertaken on the CBCT images, were linear and were made to the nearest 0.01mm. All radiographic examinations were performed by the same trained technician at the same scanner equipment (Soredex Scanora 3D). The 3D scans were taken in large field of view (FOV: 75 mm height x 145 mm diameter) in 90 KV, 10 mA and voxel size of 0.35 mm. The CBCT data were exported and analysed through DICOM viewer OnDemand3d (Cybermed Inc, Daejeon, South Korea).
In consideration of the previous literature it would be more appropriate to carry out a reorientation of the Dicom files from the Cone-Beam at T0 and a Voxel Based Superimposition with the subsequent timing, before measurements.
It is recommended to reorient the dicom files of the skull with previously validated protocols:
- PMID: 35691961 PMCID: PMC9189077 DOI: 10.1186/S40510-022-00413-8
- PMID: 23623785 DOI: 10.1016/J.IJOM.2013.03,007
In this study we did not perform reconstructions of the airway from DICOM files and we don’t have volumetric data from a software because we didn’t have one available.
Scientific English in the text should be improved.
I made English language spelling corrections.
Round 2
Reviewer 1 Report
The authors substantially improved the manuscript quality.
Reviewer 2 Report
No other operation required, the article can be published.